# Cryostructuring of Polymeric Systems. 57. Spongy Wide-Porous Cryogels Based on the Proteins of Blood Serum: Preparation, Properties and Application as the Carriers of Peptide Bioregulators

**DOI:** 10.3390/gels6040050

**Published:** 2020-12-14

**Authors:** Egor V. Sidorskii, Mikhail S. Krasnov, Viktoria P. Yamskova, Vladimir I. Lozinsky

**Affiliations:** 1A.N. Nesmeyanov Institute of Organoelement Compounds, Russian Academy of Sciences, Vavilov Street, 28, 119991 Moscow, Russia; sneegr@gmail.com (E.V.S.); embrmsk@mail.ru (M.S.K.); 2Institute of Bioregulation Problems, Leninskii Avenue 45, 119334 Moscow, Russia; Yamskova-vp@yandex.ru

**Keywords:** blood-serum-proteins-derived cryogels, delivery systems, peptide bioregulators, in vitro biotesting

## Abstract

Wide-pore proteinaceous freeze–thaw spongy gels were synthesized via the cryotropic gelation technique using the bovine blood serum or its diluted solutions as the protein-containing precursors. The feed systems also included the denaturant (urea) and the thiol-reductant (cysteine). The gel-fraction yield decreased and the swelling degree of the walls of macropores in such heterophase matrices increased with decreasing the initial protein concentration. The optimum freezing temperature was found to be within a rather narrow range from −15 to −20 °C. In this case, the average size of the macropores in the resultant cryogels was 90–110 μm. The suitability of such soft wide-pore gel materials for the application as the carriers of peptide bioregulators was demonstrated in the in vitro experiments, when the posterior segments of the *Pleurodeles waltl* adult newts’ eyes were used as a model biological target. It was shown that a statistically reliable protective effect on the state of the sclera, vascular membrane and retinal pigment epithelium, as well as on the viability of fibroblasts, was inherent in the proteinaceous cryogels loaded with the peptide bioregulator (Viophtan-5™) isolated from the bovine eye sclera.

## 1. Introduction

A large number of various gel materials based on substances of a protein nature were used for a long time and are currently employed for biomedical purposes [1,2,3,4,5,6,7,8]. The key features of these gel matrices are their biocompatibility and, in principle, biodegradability [9,10]. Specific groups of the polypeptide- and protein-composed gels are those prepared in moderately-frozen systems via the so-called cryotropic gelation processes [11,12,13,14,15,16,17,18,19,20,21,22] that result in the formation of proteinaceous cryogels (e.g., see [10,23,24,25,26,27,28,29,30,31,32]). Among the latter gel matrices of certain applied interest are biodegradable wide-porous cryogels based on individual blood proteins or their natural (plasma, serum, cell lysate) and artificial mixtures [10,11,22,23,24,25,31,33,34,35,36,37]. Such cryogels can be prepared in accordance with two major schemes:

(i) by the 3D coupling of the protein macromolecules with the aid of auxiliary cross-linking reagents added to the precursor solution directly prior to its cryogenic processing [10,11,22,23,24,25,29,30,31,34];

(ii) by inducing the unfolding of the protein globules with simultaneous interchain cross-linking through the thiol–disulfide exchange reactions, the process also caused by the denaturants introduced into the feed system before its freeze–thaw treatment [10,22,25,33,36,38].

In the latter case, no foreign inclusions are inserted into the chemical structure of the resultant cryogels that consist only of the amino acid monomeric units, and the covalent nodes within the 3D network are disulfide bridges, i.e., cystine-belonging structures [33]. Therefore, such proteinaceous cryogels are non-toxic, upon biodegradation they decompose into harmless natural amino acids, and the wide-pore morphology of these cryogels provides diverse opportunities to load similar sponges with the necessary medicines, i.e., to employ such biopolymeric matrices as drug delivery systems. In particular, cryogels derived from the blood serum albumin have successfully been examined as carriers of antibiotics and bactericides [22,38,39] in the respective experiments in vitro and in vivo. In turn, cryogels constructed on the basis of total blood plasma proteins have been applied as covers on full-thickness excision wounds [36]. In all the above-indicated in vivo cases, no undesirable inflammatory effects were observed, i.e., such blood-protein-based sponges were non-toxic and biocompatible.

The possibility to implement similar cryogels as the implantable delivery carriers of special biologically-active proteins and peptides is of independent significance since the latter compounds are used for the treatment of various diseases, among which are cancer, diabetes, bone fracture, orthopedic problems, etc. [40,41]. Of the peptide/protein bioregulators of this group, many of them are glycosilated derivatives and consist of peptide–protein complexes with similar physico–chemical properties exhibiting close biological activity [42,43]. The first studies on the use of similar bioregularors loaded in serum-albumin-derived cryogels for the induction of bone tissue regeneration have shown rather promising potential of such delivery systems [44]. In the respective experiments this type bioregulator-loaded sponge has been implanted into the model bone defects in rats thus inducing the restoration of dense bone tissue, the formation of bone marrow, and the recovery of osteons. Hence, these bioartificial implants were effective as osteoconductors due to their impact on the osteoblast precursors.

Therefore, it was of interest and of importance to also reveal the possibilities of other types of blood-protein-derived cryogels to serve as carriers for various peptide bioregulators with respect not only to bone tissue repair, but also for the induction of the regeneration of other biological tissues. The elucidation of this problem was the general goal of the present study.

To this end, sponge-like proteinaceous cryogels were prepared, originating from the total proteins of bovine blood serum; then, some physico–chemical characteristics of such materials and their macroporous morphology were evaluated followed by loading of the cryogels with the bioregulators isolated either from the serum of bovine blood [45] or from the sclera of bovine eyes [46]. The subsequent biological testing of the delivery matrices thus obtained was accomplished using the earlier developed biological model [47]. This model is a short-term cultivation of the posterior part of a newt’s eye, including the scleral, vascular and retinal membranes of the eye without the anterior segment and vitreous. This approach allows the study of the maintenance of cell viability and the state of the extracellular matrix in cultured tissues of an eye, as well as tracking their response for the action of added bioregulators.

## 2. Results and Discussion

### 2.1. Synthesis of Proteinaceous Cryogels and Their Characterization

The wide-pore cryogels based on the blood serum proteins were prepared by a procedure similar to the method we applied earlier for the synthesis of the individual protein-derived (namely, serum albumin) cryogels [33,38]. As was pointed out in the Introduction, the mechanism of such gel-formation includes the unfolding of protein globules caused by the added denaturant (urea, guanidine hydrochloride, etc.) with a simultaneous relay-race sequence of the interchain thiol–disulfide exchange reactions initiated by the small amount of a thiol-bearing reductant (e.g., cysteine) [33]. However, in the present study, when the feed solution contained the multicomponent precursors, namely, blood serum which consists of many different proteins, finding both the optimal their initial concentration and the cryogenic processing temperature for obtaining the resultant cryogels most suitable for the subsequent biological experiments was required. Therefore, these parameters, protein concentration and freezing temperature, were the tested variables in the course of such an “optimization”. Herewith, the values of the gel-fraction yield (*Y*) were used as the indicators of the gelation efficiency. The values of the swelling extent (*S*_w/w_) were measured as indicators of the cross-linking “level” of the 3D network within the polymeric phase (the walls of macropores) for these heterophase gel matrices, since the lower is such swelling, the higher is the specific amount of the cross-links inside this phase [11].

The choice of the cryogenic processing temperature interval was stipulated by the following reasons. At the negative temperatures as high as −15 °C, the feed solutions did very frequently not freeze at all because of the supercooling effects. In turn, when the process was carried out at the temperatures as low as −25 °C, the *Y* values were very small, especially for the systems with the protein concentration of 50.0 and 32.5 mg/mL. The diagrams in Figure 1 show how the efficiency of the cryotropic gel-formation at three minus temperatures was dependent on the protein concentration in the precursor solution.

It was found that in these systems the highest efficiency of the gel-formation was achieved at 15 °C, and lowering the cryogenic processing temperature to −20 °C or, moreover, to −25 °C resulted in a decrease in the gel-fraction yield from 81–93% to 78–80%, and further to 62–64%. With respect to this trend, it was virtually the same one as in the cases of proteinaceous cryogels prepared on the basis of serum albumin only [33], thus demonstrating a common character of the effect for both types of the precursor systems, individual albumin and a mixture of serum proteins. For these two systems, analogous tendency was observed with regard to the influence of the starting protein concentration on the *Y* values. Over the concentration ranges of the protein precursors used in this study, i.e., 37.5–75.0 mg/mL, and 30–50 mg/mL for the individual serum albumin (Table 1 in [33]), the difference in the gel-formation efficiency was rather small, when the cryogels were formed at −20 and −25 °C, and somewhat larger for the samples formed at −15 °C (Figure 1). In the latter case, the evidently lower gel-fraction yield values were characteristic of the samples prepared on the basis of the feed solutions with the total protein concentration of 37.5 mg/mL; hence, further decrease in the content of these macromolecular precursors in the feed systems was unreasonable.

As for the influence of initial protein concentration and conditions of cryogenic processing on the swelling characteristics of the polymeric phase in the resultant wide-pore cryogels, it was found that the *S*_w/w_ values of the equi-concentrated samples formed at −15 and −20 °C were very close, whereas the pore walls in the spongy samples prepared at −25 °C swelled considerably stronger in comparison with the former ones (Figure 2). These data testify that 3D polymeric networks of the proteinaceous cryogels formed at the lowest temperature were much less cross-linked compared to the samples synthesized at −20 and −15 °C. With that, the cross-linking extent also depended on the initial protein concentration: the higher this parameter was, the lower the respective *S*_w/w_ values were, i.e., the cross-linking density of the polymeric network was higher. The conclusion from the data on the optimum synthesis temperature range (Figure 1 and Figure 2) for such gel-forming system is rather evident: this range is relatively narrow; it lies in the vicinity from about −15 to around −20 °C.

It is also well-recognized that the freezing temperature is the key parameter that dominantly affects the size, the shape and the amount of the pores in various polymeric cryogels [11,12,13,14,15,22,48,49,50]. It is so since the lower is such a temperature that the porogen particles, i.e., the polycrystals of the frost-bound solvent, are smaller, but this occurs provided that the freezing process is not sophisticated by the pronounced supercooling effects.

We evaluated the macroporous morphology of the blood-serum-proteins-derived cryogels formed at different freezing temperatures. The images of the respective samples are shown in Figure 3.

In general, the microtexture of these spongy cryogels is typical for other known wide-pore chemically cross-linked cryogels formed via the moderate freezing of the solutions (aqueous, as a rule) that contain the relevant precursors [11,12,13,14,15,22,32,48,49,51,52,53,54]. Dark strand-like structural elements in the images of Figure 3 are the methylene-blue-stained polymeric walls of macropores; the lighter regions of the images are the macropores themselves. The ImageJ-assisted measurements of the average size of macropores gave the following values for these samples: 108.3 ± 13.8 μm (**a**), 101.4 ± 13.2 μm (**b**) and 95.0 ± 11.4 μm (**c**), i.e., it was a normal trend: at a lower freezing temperature somewhat smaller pores were formed inside the cryogel’s bulk. Taking into account the data on the gel-fraction yield (Figure 1) and swelling characteristics (Figure 2) in combination with the macroporosity properties of similar blood-serum-proteins-derived cryogels (Figure 3), one can conclude that such conditions for their synthesis as the initial concentration of protein substances of about 50 mg/mL and a freezing temperature in the vicinity of −20 °C are, most probably, the optimum conditions for the reproducible preparation of such gel materials.

Since the character of these dependences is virtually the same as for the cryogels based on individual serum albumin [33], which is the main component by weight among the serum proteins [55], it was reasonable to assume that the albumin chains are also the major constituent of the polymeric network of the cryogels prepared in the present study.

However, what about the other protein components of the initial blood serum?

In order to clarify this problem, we analyzed, via the gel-electrophoresis technique, the composition of the respective cryogels after their dissolution in the sodium dodecyl sulfate/dithiotreitol medium (see Section 3.2.4). The results thus obtained are illustrated by the poly(acrylamide) gel (PAAG)-electrophoregrams in Figure 4.

Large amounts of different proteins are well-known to constitute in the blood serum composition, the major fraction being albumins; in several times smaller amounts various globulins are present; other proteinaceous components are present in minor fractions [56,57,58]. Depending on the serum source, i.e., the type of organism of the donor, its age and health, the fraction ratio is varied. It concerns also the amount of the iso-forms of particular proteins. For instance, it is the case in the tracks 1 and 2 in Figure 4 when two fractions of serum albumin inherent in human or animal blood of “persons” possessing the so-called bisalbuminemia disorder are present [59].

It was found that not only albumin polypeptide chains, but also those belonging to certain other proteins of the blood serum (immuglobulins, transferrin, α-1 and α-2 globulins, in particular) were built in the spatial network of the proteinaceous cryogels formed in frozen systems containing a sum of serum proteins, denaturant (urea) and thiol reductant (cysteine). Of course, Figure 4 shows the qualitative data; the quantification of the ratio between different proteinaceous components in the cryogel matter is the target of our further study. Nonetheless, this electrophoregram testifies to a common mechanism of the globular serum proteins’ participation in the formation of the disulfide-crosslinked polymeric networks of similar proteinaceous cryogels. In addition, these results allow the supposition that certain components of the resultant blood-serum-proteins-derived matrices could exert favorable or unfavorable influence on the properties of such materials upon their use as the carriers of the respective bioregulators. The elucidation of the latter supposition was obtained during our subsequent in vitro experiments using a model biological target.

### 2.2. Testing of Proteinaceous Cryogels as the Carriers of Peptide Bioregulators

In the Introduction, it was indicated that the wide-pore proteinaceous cryogels could be of applied biomedical interest as biocompatible delivery carriers for various peptide/protein bioregulators. Therefore, in order to confirm this statement we carried our certain biotesting experiments, in which the blood-serum-proteins-derived cryogels were used as such carriers, while the commercial biological products Viophtan-1™ and Viophtan-5™ were employed as the peptide bioregulators (see Section 3.1), and the posterior segments of the adult newts’ eyes were the biological matter under testing, i.e., the latter ones act as the model biological targets.

The following changes in the state of tissues and cells of the posterior segment of the newt’s eye were observed in the course of culturing of these systems that were subdivided in to six experimental groups indicated in Section 3.2.7.

(i) Reference samples from Group 1 (Figure 5a):

When the newt eye segments were incubated into the serum-free culture medium without any foreign additives and enclosures (spongy cryogels), both the detachment of the retina (**r**) from the retinal pigment epithelium (**pe**) layer and the detachment of this layer from the vascular membrane occurred. In the layer of the retinal pigment epithelium itself, the pigment shifted to the apical side thus indicating the cells’ instability within this layer and their de-differentiation. The degradation and damage of neurons in the retina were also observed. The signs of the beginning of tissue degradation in the sclera were detected. The latter changes were expressed in the stratification of collagen fibers with the formation of large cavities (**c**) between them, as well as in a decrease in the number of fibroblasts (**f**) per a square unit of the microscopy image (***1*** in Figure 6). These data point to the processes of the beginning of degradation of all tissues and cells of the posterior part of the eye under the cultivation conditions used during the 72-h-long experiments.

(ii) Reference samples from Group 2 (Figure 5b):

In the case of the newt eye segments incubation onto the non-loaded proteinaceous cryogel supports in the serum-free culture medium without any foreign soluble additives, i.e., bioregulators, a somewhat “better” (with respect of the eye tissues state) pattern was observed in comparison to the samples of Group 1. Although the detachment of the retina (**r**) from the retinal pigment epithelium (**pe**) layer also occurred, the outer segments of photoreceptor cells and other retinal neurons were less damaged than those in the samples of Group 1. The pigment in the retinal pigment epithelium layer also tended to shift to the apical side, but not so pronouncedly as for the Group 1 samples. The vascular shell was dense. In the sclera, there were also the elements of tissue degradation that were exhibited in the stratification of the collagen fibers and the formation of cavities (**c**), as well as in a decrease in the number (***2*** in Figure 6) of fibroblasts (**f** in Figure 5). It could be suggested that, in this case, some tissues did not degrade as much as in Group 1, but the state of the scleral membrane itself and its cells also underwent certain degradation processes during the same cultivation time.

(iii) Reference samples from Group 3 (Figure 5c):

When the newt eye segments were incubated in the culture medium with the additives of the Viophtan-1 bioregulator, but without the spongy cryogel supports, the detachment of the retina (**r**) from the retinal pigment epithelium (**pe**) was observed as well, and the pigment shifted in the retinal pigment epithelium (**pe**) layer to the apical side. The vascular shell was dense without any signs of damage. The exhibition of the tissue degradation was detected in the sclera to a lesser extent than for the samples from Groups 1 and 2. This effect was expressed in a smaller size of the cavities (**c**) between the collagen fibers (**f**), as well as in a larger number of fibroblasts (***3*** in Figure 6). Such results indicate that the addition of the Viphtan-1 bioregulator to the culture medium exerted a slight protective influence on the eye tissues in this biological model, mainly on the scleral membrane.

(iv) Experimental samples from Group 4 (Figure 5d):

In the case of the newt eye segment incubation onto the spongy cryogel support loaded with the Viophtan-1 bioregulator, the detachment of the retina (**r**) from the retinal pigment epithelium (**pe**) was also observed, and the pigment in the retinal pigment epithelium layer was also shifted to the apical side. The vascular shell was in a good condition, dense, without signs of degradation. In the sclera, the elements of tissue degradation were observed being expressed in the stratification of collagen fibers and in the formation of cavities (**c**). The number of fibroblasts (**f**) turned out to be higher (by about 1.5-fold) compared to the reference samples from Groups 1–3 (***4*** in Figure 6). We assume that the latter result could be related to the favorable influence of the proteinaceous support *per se*. With that, the effects of the adhesive factor of the spongy support together with the presence of the Vioftan-1 could be summed up.

(v) Reference samples from Group 5 (Figure 5e):

When the newt eye segments were incubated without the spongy cryogel supports in the culture medium with the additives of Viophtan-5, the sclera-originated bioregular, the best visual picture of the histological state of tissues after cultivation under the selected conditions was observed. This was manifested in the fact that the detachment of the retina (**r**) from the retinal pigment epithelium (**pe**) layer was practically not observed throughout the posterior part of the eye; only minor detachments were visible in some areas. The pigment in the layer of the retinal pigment epithelium did not shift to the apical side of the cells and was distributed evenly and compactly in the cells of the retinal pigment epithelium. Such a result indicates a stable state of the cellular differentiation of these cells. The vascular shell was preserved and dense, without signs of degradation. In the sclera, there were no pronounced changes in tissue degradation, as was detected for the samples in Groups 1–4. The collagen fibers were compactly located with no formation of large cavities (**c**). The number of fibroblasts (**f** in Figure 5) turned out to be significantly higher than in the Groups 1–4 (***5*** in Figure 6). The comparison of the data for the samples belonging to the Groups 3 and 5, both without the spongy cryogel supports, but containing the peptide bioregulators from different biological sources (blood serum and eye sclera), testifies to the expressed tissue specificity of the sclera-isolated bioregulator relatively to the eye tissue [60].

(vi) Experimental samples from the Group 6 (Figure 5f):

In the case of the newt eye segment incubation onto the spongy cryogel support loaded with the Viophtan-5 bioregulator, the following histological picture was observed. Some minor detachment of retina (**r**) from the retinal pigment epithelium (**pe**) layer was registered, while the retinal pigment epithelium layer itself was not detached from the vascular membrane. Pigment displacement in the retinal pigment epithelium layer was not virtually observed, thus indicating the stabilization of the differentiated state of cells in this layer. The sclera contained minor degradation elements, namely, minor cavities (**c**) and stratification of collagen fibers. These effects were less pronounced than in the reference samples from the Groups 1–3 and in the experimental samples from the Group 4, which contained a blood serum bioregulator. At the same time, the effects were more pronounced than for the reference samples belonging to the Group 5. The vascular shell was smooth without pronounced elements of tissue degradation. The amount of fibroblasts (**f**) was higher than in all other groups and approximately twice as large compared to reference Group 1. The positive effect of the Vioftan-5-loaded cryogel should be noted, since the statistically significant number of fibroblasts per unit area was found. The amount of these cells serves as a quantitative indicator, and the condition of tissues at a qualitative level, especially the scleral membrane, did not differ much in Groups 5 and 6, but was significantly better than in all the other groups. The cultivation process is an unfavorable environment in which fibroblasts gradually die over time. The more fibroblasts observed in the sclera during cultivation, the more factors added to the culture medium have the effect of maintaining the viability of fibroblasts. Therefore, we concluded that the combined composition of cryogel and bioregulator has a positive effect based on the quantitative criterion because the fibroblasts in the sclera play a key role in maintaining the metabolism and synthesis of collagen fibers that give strength to the scleral membrane. Hence, these data indicate the ability of the Vioftan-5-loaded spongy proteinaceous cryogel composite systems to prevent the development of degenerative processes in the sclera membrane and to maintain adhesive interactions between the sclera and adjacent tissues.

In the native part of the eye, before any incubation (Figure 5g), close adhesive interactions can be observed between the tissues of the retina, choroid and sclera of the eye. Moreover, the pigment in the layer of the pigment epithelium is distributed evenly and densely. In the scleral shell of the eye itself in the presence of dense collagen fibers, the absence of cavities occurs.

In general, the juxtaposition of the information presented in Figure 5 and Figure 6 allows one to conclude that exactly the Viophtan-5 peptide bioregulator exhibits a statistically reliable protective effect on the state of the sclera, vascular membrane and retinal pigment epithelium as well as preserving the viability of fibroblasts which are responsible for maintaining the scleral membrane [61,62]. With that, a sponge-like morphology of the blood-serum-derived cryogel support provides enhanced delivery of such a tissue-specific bioregulator to the biological target of interest, the posterior segment of the adult newts’ eye in this particular case.

## 3. Experimental

### 3.1. Materials

The following substances were used in the experiments without additional purification: bovine blood serum (lot. #015BS247; Biosera, Nuaille, France); bioregulators Viophtan-1™ (source—bovine blood serum) and Viophtan-5™ (source—sclera of bovine eyes) (both biological agents were obtained from the LLC Institute of Bioregulation Problems, Moscow, Russian Federation); Medium 199 (Gibco, Paisley, UK); urea (Sigma, St. Louis, MO, USA), cysteine, eosin, hematoxylin (all—DIA-M, Moscow, Russian Federation); methylene blue (Merck GmbH, Darmstadt, Germany); sodium dodecyl sulfate (SDS) and Coomassie R250 dye (Serva, Heidelberg, Germany); 1,4-dithiotreitol (DTT), xylene and acetone (all—Panreac, Barcelona, Spain); ethyl alcohol (CHIMMED, Moscow, Russian Federation). All aqueous solutions were prepared using Milli-Q quality water.

### 3.2. Methods

#### 3.2.1. Synthesis of Wide-Porous Cryogels Based on the Blood Serum Proteins

The target proteinaceous cryogels were prepared using either total bovine blood serum with a sum protein concentration of 75.0 mg/mL or the solutions formed after the serum dilution with water up to the following protein concentrations: 62.5, 50.0 and 37.5 mg/mL. Then, the required amount of urea was added to each solution in order to reach a urea concentration of 1.5 mol/L; the solutions were cooled in an ice bath, followed by the dissolution of the required amount of the cysteine to reach a concentration of 0.01 mol/L. The reaction mixture thus prepared was poured in 1.5-mL portions into plastic Petri dishes (an internal diameter of 38 mm) that were sealed and placed in the chamber of a programmable cryostat F-32 ME (Julabo, Seelbach, Germany) with a pre-set sub-zero temperature, where the samples were incubated for 24 h. The samples were then removed from the cryostat, thawed for 30 min followed by rinsing from the sol-fraction in pure water (4 × 50 mL per a sample).

#### 3.2.2. Characterization of the Synthesized Cryogels

The gel-fraction yield and the degree of swelling of cryogels based on serum proteins were measured gravimetrically as follows. To do this, a spongy sample swollen in water was placed on a glass filter and the free solvent was removed with a water pump vacuum for 1 min under a load of 170 g, and then the sample was kept for 4 min, also under a vacuum, but without load. The wet sponge pressed in this way was weighed and dried to a constant weight at 105 °C in a SNOL 24/200 oven (AB UtenosElektrotechnika, Vilnius, Lithuania).

The yield of the gel fraction (Y) was calculated from the ratio:*Y* = (*m*_dry_/*m*_theor_) × 100%,
where *m*_dry_ is the weight of the dried sample, and *m*_theor_ is the weight of the protein under the condition of 100% gelation yield.

The degree of swelling by weight (*S*_w/w_) of the polymeric phase (the walls of macropores) of the wide-pore samples was calculated by the formula:*S*_w/w_ = (*m*_wet_ − *m*_dry_)/*m*_dry_ (g H_2_O per 1 g of dry polymer),
where *m*_wet_ is the weight of the “pressed-off” wet sample.

#### 3.2.3. Microstructure of Proteinaceous Cryogels

Cryogel samples for the microscopic studies were formed as 1 mm-thick disks; the cryogenic processing temperature was either −15, or −20, or −25 °C. After the preparation and rinsing from the sol-fraction, the discs were stained with 0.125 mM aqueous solution of methylene blue dye for 1 min, followed by thorough washing with water. The macroporous morphology of the resultant samples was studied using an SMZ1000 optical stereomicroscope (Nikon, Tokyo, Japan) equipped with the digital image recording MMC-50C-M system (MMCSoft, St.Petersburg, Russian).

#### 3.2.4. Component Composition of the Cryogels Based on Blood Serum Proteins

The component composition of the synthesized proteinaceous cryogels was evaluated with the aid of the electrophoresis in poly(acrylamide) gel (PAAG) after the overnight dissolution of the respective cryogel samples in the 1% (*w*/*v*) SDS solution containing DTT (0.05 mol/L). Electrophoresis according to Laemmli’s procedure [63] was carried out using the vertical PAAG plates of 1 mm thickness; 12.5% and 4% were the concentrating and the separating gels, respectively. The kit of marker proteins 2–250 kDa (Bio-Rad, Hercules, CA, USA) was used as the standards for the determination of molecular weights. The process was performed at a constant current of 20–30 mA for 1 h followed by standard fixing with an iso-propanol/acetic acid solution and staining with a Coomassie R250 dye.

#### 3.2.5. Preparation of the Bioregulator-Loaded Cryogels and Reference Samples

The loading of the blood-serum-proteins-based cryogels with the respective bioregolators was performed as follows. First, a free liquid was removed for 3 min on a glass filter under a vacuum from the spongy samples swollen in water, then the resulting “squeezed out” wet cryogel was put in an aqueous solution of the Viophtan-1 or Viophtan-5 bioregulator to saturate the sponge carrier. The swollen cryogel samples were incubated in these solutions for 4 h at 15 °C, after which they were frozen at −20 °C and freeze-dried. The reference sponges that did not contain any bioregulators were, after rinsing with water, also frozen and freeze-dried.

#### 3.2.6. Preparation of Biological Samples

The studies were performed using the eyes of the *Pleurodeles waltl* adult newts of both sexes grown in the aquarium of the N.K.Koltsov Institute of Developmental Biology, Russian Academy of Sciences. Nine animals (18 eyes) were used in each cultivation experiment. The newts were anesthetized with a 2% solution of ethyl urethane in amphibian saline (0.65% NaCl). After anesthesia, the animals’ heads were rinsed with 70% ethanol, and the eyes were enucleated under standard laboratory lighting. Isolated eyes were rinsed and stored in the sterile 35 mm Petri dishes with amphibian medium (199 medium—70% and distilled water—30%). The medium for culturing eye tissue had the following composition: 350 mL of the 199 medium, 150 mL of bidistilled water, 0.15 mL of 1.0 M HEPES buffer, 1 mL of the 1% gentamicin solution and 0.5 mL of the antibiotic/antimycotic Gibco^®^ solution (Thermo Scientific, Waltham, MA, USA). Before being introduced into the vials, the medium was cold-sterilized passing through the membrane filters of the FPE-204-030 type (Jet Biofil, Guangzhou, China), with a pore size of 0.22 μm. The test eye tissues were prepared under a binocular lens in the following sequence of manipulations: the eyes were cut along the circumference, proximal to the limb. The growth region of the retina, along with the iris, cornea, and lens, were discarded. The posterior segment of each eye, which included the retina, pigment epithelium, vasculature, and sclera, was used for the subsequent cultivation.

#### 3.2.7. Cultivation Experiments and Analysis of the Resulting Biological Samples

The following six groups of samples were involved in these experiments:

Group 1. The eye-derived segment in the cultivation medium (10 mL) without foreign additives and 0.1 mL of physiological saline; the system did not contain a cryogel sponge and bioregulators (reference samples).

Group 2. The eye-derived segment placed onto the non-loaded cryogel sponge in the cultivation medium (10 mL) with 0.1 mL of physiological saline added (reference samples).

Group 3. The eye-derived segment in the cultivation medium (10 mL) with 0.1 mL solution of a Vioftan-1 bioregulator added (source—the bovine blood serum) (peptide/protein concentration of 10^–8^ mg/mL); the system did not contain a cryogel sponge (reference samples).

Group 4. The eye-derived segment placed onto a cryogel sponge loaded with a Vioftan-1 bioregulator; the system contained 10 mL of the cultivation medium and 0.1 mL of physiological saline (experimental samples).

Group 5. The eye-derived segment in 10 mL of the cultivation medium with 0.1 mL solution of a Vioftan-5 bioregulator added (source—sclera of bovine eyes) (peptide/protein concentration of 10^–8^ mg/mL); the system did not contain a cryogel sponge (reference samples).

Group 6. The eye-derived sample placed onto the cryogel sponge loaded with a Vioftan-5 bioregulator; the system contained 10 mL of the cultivation medium and 0.1 mL of physiological saline (experimental samples).

The experiments were performed in the small vials each containing 10 mL of the serum-free cultivation medium, as well as the respective soluble additives and enclosures (Figure 7).

Cultivation was carried out at 20–22 °C for 72 h in the dark, without changing the culture medium. The state of the explants after the cultivation period was studied on a series of paraffin sections. The eye tissue segments were fixed in the 10% formalin aqueous solution for 12 h, rinsed three times with 70% ethanol, then dehydrated by the media of increased ethanol concentration, rinsed with xylene and poured into paraffin in accordance to the standard procedure [64]. Paraffin sections with a thickness of 7 microns were made using the ERM 4000 microtome (Hestion, Melbourne, Australia).

After dewaxing and hydration, these thin sections were stained with hematoxylin and eosin followed being placed under a cover glass with the addition of an adhesive liquid. A Jenaval microscope (Carl Zeiss, Jena, Germany) was used to the view histological sections. Assessment of the number of viable fibroblasts on histological sections was performed using the ImageJ program, estimating the number of fibroblasts in the sclera tissue relative to the area of the entire section. At least 30 s were examined for each experimental point. The results were processed according to the Mann–Whitney criterion [65]. 

## 4. Conclusions

The wide-pore proteinaceous cryogels are of interest as biocompatible materials for biomedical purposes. In this study we synthesized a series of similar spongy cryogels based on the protein mixture being the constituents of the bovine blood serum. The preparation technique was grounded in the known phenomenon that the solutions of such globular proteins in the presence of added denaturants like urea and the thiol-reductants like cysteine are capable of the cryotropic gel-formation. Now, it was shown that the gel-fraction yield decreased and the swelling degree of the walls of macropores in the prepared matrices increased with decreasing the initial concentration of the blood serum proteins. The optimum freezing temperature was found to be within a rather narrow range from −15 to −20 °C. The average size of the macropores in the resultant cryogels was 90–110 μm, and polypeptide chains belonging not only to albumin, but to some other serum proteins (immunoglobulins, transferrin and various globulins, in particular) present in the composition of the 3D polymeric network in the cryogels were thus obtained. Subsequent evaluative experiments on the application of such soft wide-pore gel materials as the carriers of peptide bioregulators have demonstrated the promising possibilities of the given approach. When the posterior segments of the *Pleurodeles waltl* newts’ eyes were used as a model biological targets, a statistically reliable protective effect on the state of the sclera, vascular membrane and retinal pigment epithelium as well as on the viability of fibroblasts was found to be inherent in the proteinaceous cryogels loaded with the peptide bioregulator (Viophtan-5™) isolated from the bovine eyes’ sclera. To the best of our knowledge, such studies have not been performed earlier, and the results obtained in the present study are thought to be prospective in view of the applied biomedical potential of similar drug delivery systems consisting of blood-proteins-based cryogel-type carriers and various peptide bioregulators. At present, in vivo biotesting of similar systems is in progress in cooperation with professionals within the biomedical fields.

## Figures and Tables

**Figure 1 gels-06-00050-f001:**
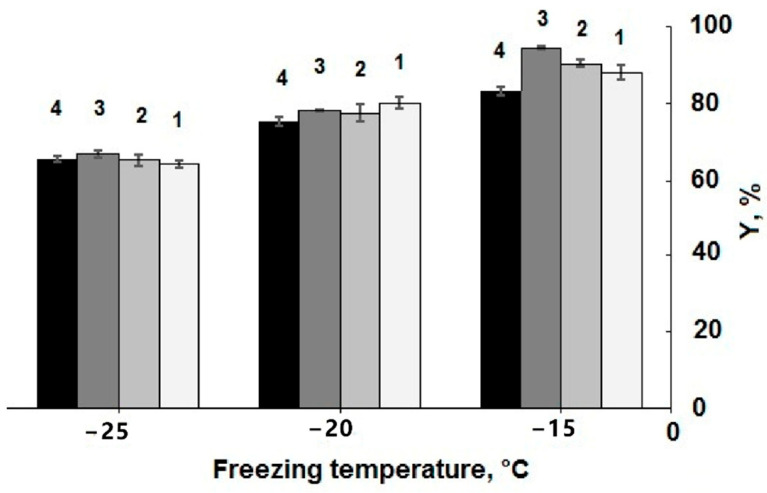
Values of the gel-fraction yield (*Y*) for the cryogels prepared from the initial solutions with different blood serum proteins concentration (75.0 (**1**), 62.5 (**2**), 50.0 (**3**), and 37.5 (**4**) mg/mL) by their freezing at −15, −20, and −25 °C.

**Figure 2 gels-06-00050-f002:**
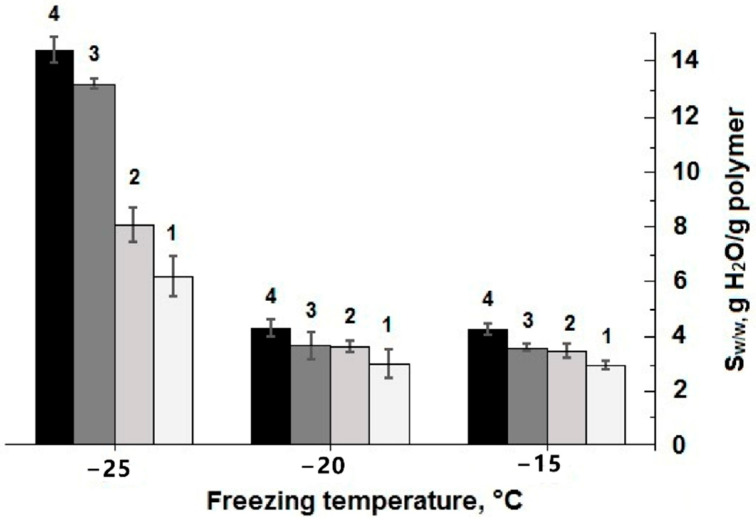
Swelling degree (*S*_w/w_) of the cryogels prepared from the initial solutions with different blood serum proteins concentrations (75.0 (**1**), 62.5 (**2**), 50.0 (**3**), and 37.5 (**4**) mg/mL) by their freezing at −15, −20, and −25 °C.

**Figure 3 gels-06-00050-f003:**
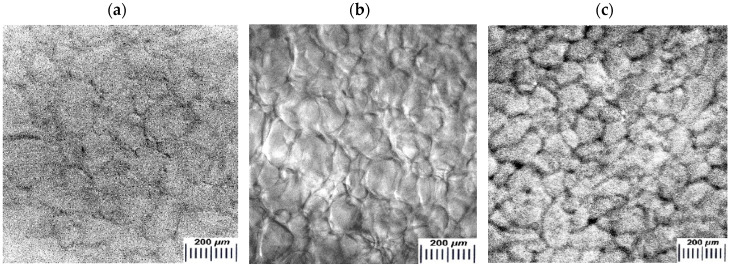
Micrographs (optical stereo microscope) of the water-swollen proteinaceous cryogels formed at −15 (**a**), −20 (**b**) and −30 °C (**c**) (the proteins concentration in the initial solutions was 50 mg/mL for all the samples).

**Figure 4 gels-06-00050-f004:**
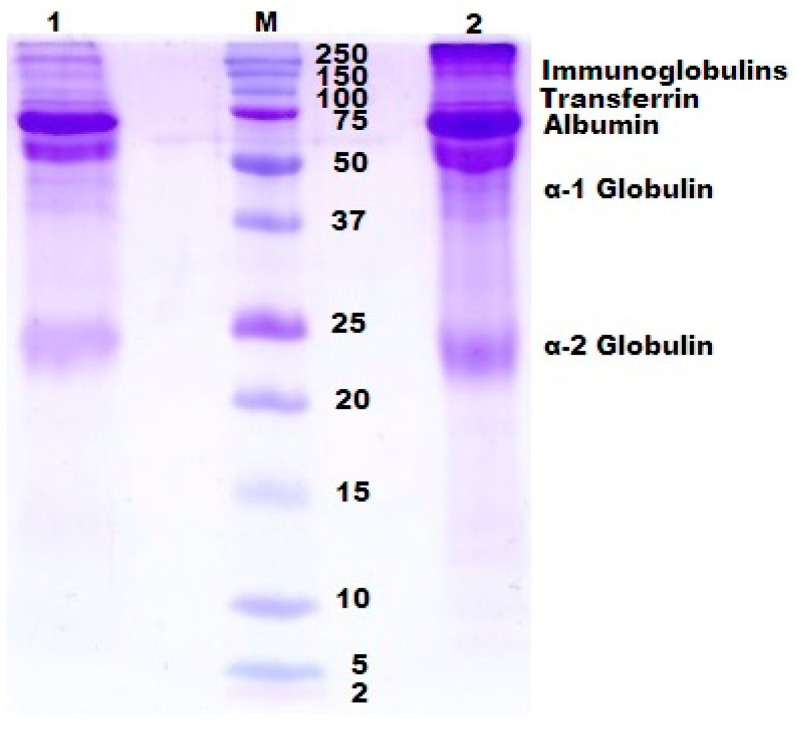
Sodium dodecyl sulfate (SDS)-1,4-dithiotreitol (DTT)-poly(acrylamide) gel (PAAG) electrophoregram of the liquid obtained as a result of the SDS-DTT-dissolution of a proteinaceous cryogel (track **1**), the molecular weight markers (track **M**) and bovine blood serum (track **2**). Experimental peculiarities: the cryogel was prepared on the basis of bovine blood serum diluted until it reached the protein concentration of 50 mg/mL; urea (1.5 mol/L) and cysteine (0.01 mol/L) were also added to the feed system; the freezing temperature was −20 °C, frozen storage duration—24 h; the PAAG-loaded samples **1** and **2** contained 10 micrograms of the proteins.

**Figure 5 gels-06-00050-f005:**
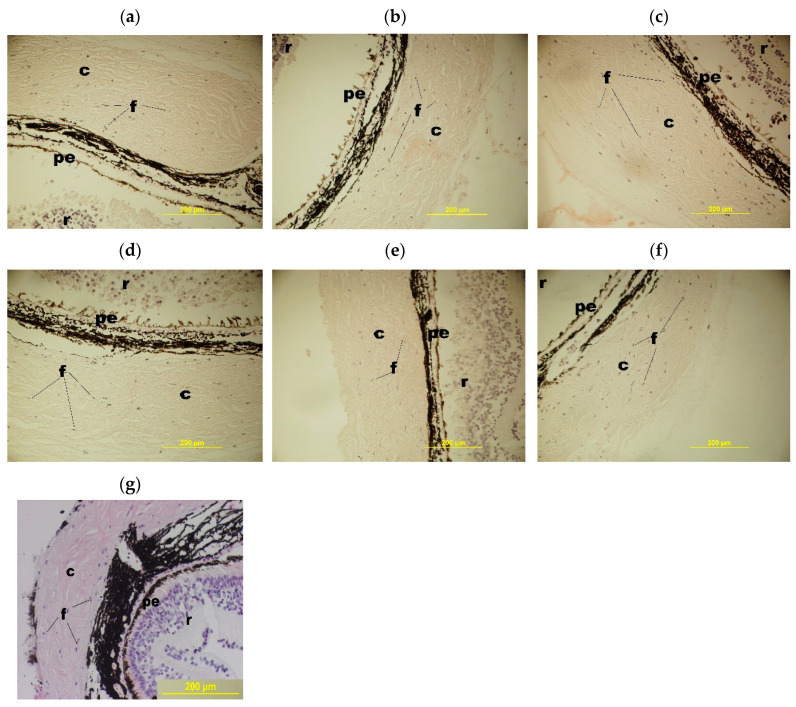
Histological micrographs of the cross-sections of the posterior part of the newt *Pl. waltl* eye after their 72-h-long organotypic cultivation. The samples were taken from the following reference and experimental Groups: 1 (**a**), 2 (**b**), 3 (**c**), 4 (**d**), 5 (**e**) and 6 (**f**); image (**g**) is the histological micrograph of the cross-sections of the posterior part of the newt *Pl. waltl* eye before any incubation (abbreviations in the micrographs: **c**—cavity, **f**—fibroblasts, **pe**—pigmented epithelium, **r**—retina).

**Figure 6 gels-06-00050-f006:**
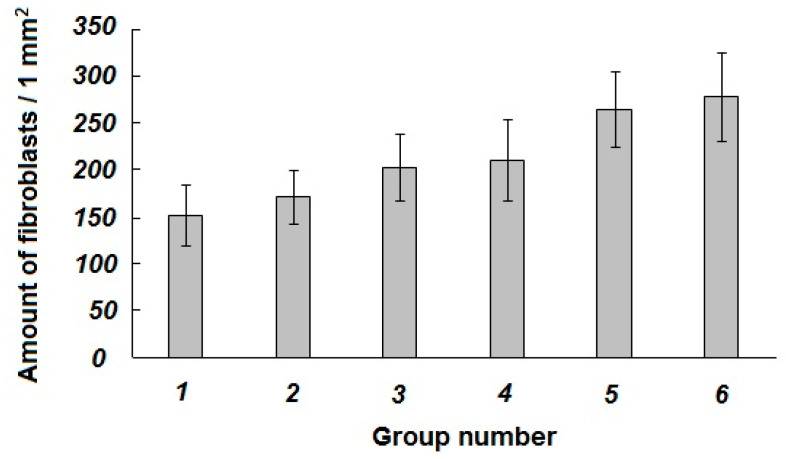
Specific amount of fibroblasts in the *Pl. waltl* newt’s eye sclera tissue after the 72-h-long organotypic in vitro cultivation of the samples belonging to different reference and experimental Groups.

**Figure 7 gels-06-00050-f007:**
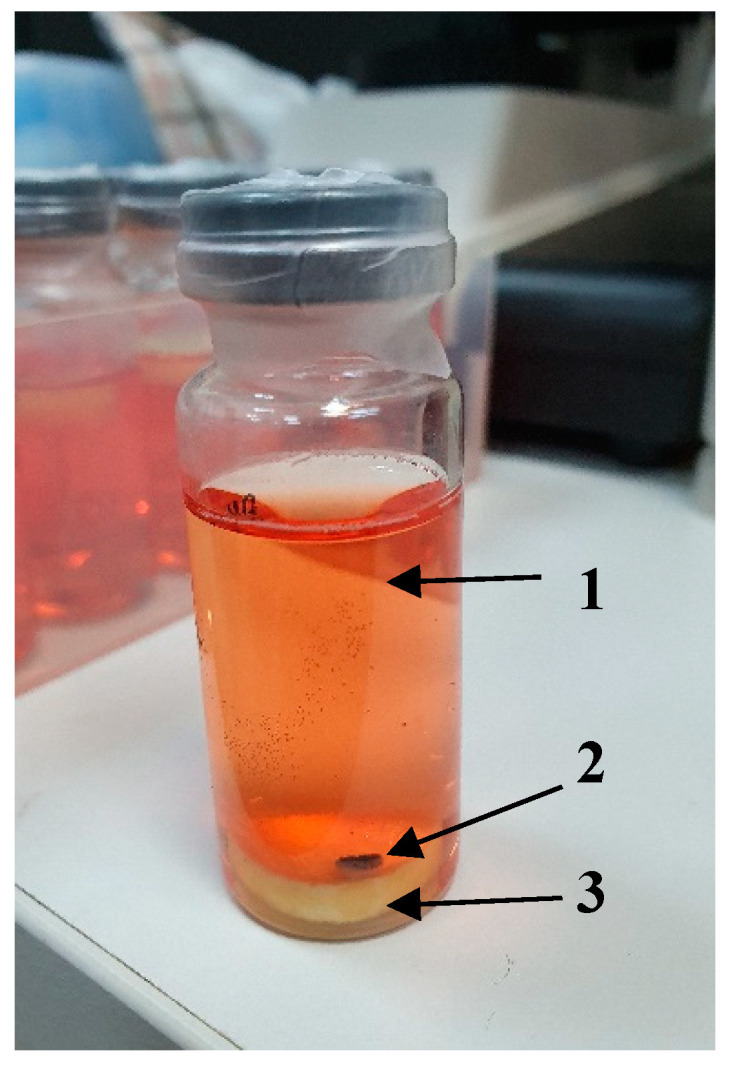
The appearance of the vial, which contains the model system of the eye-derived segment cultivation (**1**—cultivation medium; **2**—eye-derived segment; **3**—cryogel sponge).

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
