# Peer review of "Cryostructuring of Polymeric Systems. 57. Spongy Wide-Porous Cryogels Based on the Proteins of Blood Serum: Preparation, Properties and Application as the Carriers of Peptide Bioregulators"

_gels, 2020, doi:10.3390/gels6040050_

Round 1

Reviewer 1 Report

This paper can be accepted in the present form.

Author Response

Reviewer 1

This paper can be accepted in the present form.

Thank you so much for this positive opinion on our manuscript.

Reviewer 2 Report

The manuscript entitled” Cryostructuring of Polymeric Systems. 57. Spongy Wide-Porous Cryogels Based on the Proteins of Blood Serum: Preparation, Properties and Application as the Carriers of Peptide Bioregulators” by Lozinsky and his team describes the in vitro application of bioregulator in the presence of soft cryogels on the posterior segments of the Pleurodeles waltl adult newts’ eyes as a model biological target. The cryogel was synthesized using the bovine blood serum, including the denaturant (urea) and the thiol-reductant (cysteine).

Forming a macroporous structure from biodegradable and biocompatible materials is always an exciting topic for medical application. Which from my point of view, the team has a strong background in that regards. However, there are a few issues regarding the manuscript

  • The language needs to be polished. The sentences are too long in some cases and difficult to follow e.g. page 3 lines 95-99; page 4, lines 133-136 etc.
  • What was the reason for the authors to select an optical stereo microscope to demonstrate the morphology and structure of their cryogel and not SEM?
  • Figure 3, which section of the cryogel did the author choose for their analysis?
  • On page 5, lines 189-197. How can the author be sure that all the other proteins in the blood serum participated in the forming of the cryogel network and not only trapped inside the albumin? SDS-PAGE is not sufficient to conclude if the proteins actually participated in the cryogel synthesis or they were only trapped inside the wall. In either of these scenarios, the bands will be the same on the SDS-DTT-PAAG.
  • Page 6, I appreciated that the author categorized the six experiments in different groups, however I think the description of each group should be presented in the method section.
  • Figure 5, It would be helpful if the authors, presented the histological micrograph of the cross-sections of the posterior part of the newt Pl. waltl eye as a reference before any incubation.
  • The authors claim that figure 5e, gave the best results. The fair comparison of 5e and 5f has been missed in the text. From my point of view, either the authors downplayed the fact that the cryogel does not have any impact (rather negative) on the delivery of the bioregulator, Viophtan -5 or they failed to explain its effect clearly in the text.
  • The conclusion is very much like the abstract, thus needs a major revision.
  • 27% of the given references was self-citation. The topic of the cryogel is very well studied, and there many references which the authors can cite instead of their own.

Author Response

Reviewer 2

Comments and Suggestions for Authors

The manuscript entitled” Cryostructuring of Polymeric Systems. 57. Spongy Wide-Porous Cryogels Based on the Proteins of Blood Serum: Preparation, Properties and Application as the Carriers of Peptide Bioregulators” by Lozinsky and his team describes the in vitro application of bioregulator in the presence of soft cryogels on the posterior segments of the Pleurodeles waltl adult newts’ eyes as a model biological target. The cryogel was synthesized using the bovine blood serum, including the denaturant (urea) and the thiol-reductant (cysteine).

Forming a macroporous structure from biodegradable and biocompatible materials is always an exciting topic for medical application. Which from my point of view, the team has a strong background in that regards. However, there are a few issues regarding the manuscript

  • The language needs to be polished. The sentences are too long in some cases and difficult to follow e.g. page 3 lines 95-99; page 4, lines 133-136 etc.

The text has been re-edited; the changes made are indicated by the blue color in the revised manuscript.

  • What was the reason for the authors to select an optical stereo microscope to demonstrate the morphology and structure of their cryogel and not SEM?

Optical microscopy allowed studying the macroporous morphology of our cryogels in the swollen state, i.e. in the state used in “biological” experiments, whereas SEM gives the information about the structure of the samples in a dry state. Our interests were focused on the first variant.

  • Figure 3, which section of the cryogel did the author choose for their analysis?

Figure 3 shows the microphotographs of the top surfaces of the Methylene-blue-contrasted 1-mm-thick cryogel discs (see Section 3.2.3) rather than some their sections.

  • On page 5, lines 189-197. How can the author be sure that all the other proteins in the blood serum participated in the forming of the cryogel network and not only trapped inside the albumin? SDS-PAGE is not sufficient to conclude if the proteins actually participated in the cryogel synthesis or they were only trapped inside the wall. In either of these scenarios, the bands will be the same on the SDS-DTT-PAAG.

Our manuscript does not contain any statements that “all the other proteins in the blood serum participated in the forming of the cryogel network”. In order to indicate this point somewhat clearer, the following phrases have been added to the text (page 6) of the revised manuscript:

      “Of course, Fig. 4 shows the qualitative data; the quantification of the ratio between of different proteinaceous components in the cryogel matter is the target of our further studies. Nonetheless, this electrophoregram testifies a common mechanism of the globular serum proteins participation in the formation of the disulfide-crosslinked polymeric networks of similar proteinaceous cryogels.”

  • Page 6, I appreciated that the author categorized the six experiments in different groups, however I think the description of each group should be presented in the method section.

Thanks for this remark; we shifted description of these groups to the Experimental (see Section 3.2.7 in the revised manuscript).

  • Figure 5, It would be helpful if the authors, presented the histological micrograph of the cross-sections of the posterior part of the newt Pl. waltl eye as a reference before any incubation.

Such micrograph has been added as the Fig. 5g to the revised manuscript. Also, the following description of this figure has been added to the revised manuscript (indicated in blue color):

“In the native part of the eye before any incubation, close adhesive interactions are observed between the tissues of the retina, choroid and sclera of the eye. Also, the pigment in the layer of the pigment epithelium is distributed evenly and densely. In the scleral shell of the eye itself the presence of dense collagen fibers, the absence of cavities take place.”

  • The authors claim that figure 5e, gave the best results. The fair comparison of 5e and 5f has been missed in the text. From my point of view, either the authors downplayed the fact that the cryogel does not have any impact (rather negative) on the delivery of the bioregulator, Viophtan -5 or they failed to explain its effect clearly in the text.

We want to note the positive effect of the Vioftan-5-loaded cryogel, since the statistically significant number of fibroblasts per unit area was found. Quite the amount of these cells serves as a quantitative indicator, and the condition of tissues at a qualitative level, especially the scleral membrane, did not differ much in groups 5 and 6, but was significantly better than in all other groups. The cultivation process is an unfavorable environment in which fibroblasts gradually die over time. The more fibroblasts are observed in the sclera during cultivation, the more factors added to the culture medium have the effect of maintaining the viability of fibroblasts. Therefore, we concluded that the combined composition of cryogel and bioregulator has a positive effect based on the quantitative criterion because the fibroblasts in the sclera play a key role in maintaining the metabolism and synthesis of collagen fibers that give strength to the scleral membrane. This remark was also added to the revised manuscript.

  • The conclusion is very much like the abstract, thus needs a major revision.

As a rule, both the abstract of the experimental papers and the respective conclusions are close with respect of a brief description of the studies that have been made and the major obtained results. In our manuscript exactly the same case is, i.e., both this sections contain the information about the experiments performed and the data obtained. Evidently therefore neither the content of the ‘abstract’, nor of the ‘conclusions’ for our paper, did not cause the objections from the Reviewers 1 and 3. In order to point out the possible prospects of such cryogel-based systems of our interests, we, in accordance to the Reviewer 2 recommendation, inserted the additional sentence in the ‘conclusions’ of the revised manuscript (indicated by the blue color).

  • 27% of the given references was self-citation. The topic of the cryogel is very well studied, and there many references which the authors can cite instead of their own.

We made certain changes and added quotation of additional references for the publications of other authors (indicated by the blue color in the revised manuscript) so that the per cent of the self-citation related to cryogels has been reduced to ~18.5%.

Reviewer 3 Report

This manuscript studied the biocompatible materials synthesis processes using bovine blood serum. They did very good work on experimental design and analysis.

I think it will be fine to be accepted after the author revises:

1, The Y-axis in Figures 1 and 2 to keep consistency with Figure 6 and make sure the titles of "Group number" and "Freezing temperature" be centered.

2, Short the Conclusions part, you do not need to cite any references in this part and make sure your conclusion concise.

Author Response

Reviewer 3

This manuscript studied the biocompatible materials synthesis processes using bovine blood serum. They did very good work on experimental design and analysis.

Many thanks for your positive opinion on our work.

I think it will be fine to be accepted after the author revises:

1, The Y-axis in Figures 1 and 2 to keep consistency with Figure 6 and make sure the titles of "Group number" and "Freezing temperature" be centered.

Thanks for this remark; the titles of the axes have been corrected in the revised manuscript.

2, Short the Conclusions part, you do not need to cite any references in this part and make sure your conclusion concise.

The references have been omitted.